# A protocol for an assessment of the health information system in Khomas region, Namibia

**Veronika Jatileni**[1,2]**, Edward Nicol** [ID][1,3]*

**1** Division of Health Systems and Public Health, Stellenbosch University, Cape Town, South Africa, **2** Directorate of Special Programs, HIV and TB, Ministry of Health and Social Services, Windhoek Namibia, **3** Burden of Disease Research Unit, South African Medical Research Council, Cape Town, South Africa

* edward.nicol@mrc.ac.za

## Abstract

### Introduction

A robust and well-functioning Health Information System (HIS) is crucial for managing patient care, monitoring health system performance, and informing public health decisions. However, Namibia, like many developing countries, faces challenges in its HIS, such as limited financial and human resources, knowledge gaps, inadequate infrastructure, and behavioural barriers such as resistance to adopting new systems and a lack of supportive policies. Previous studies have not shown significant improvements since 2012. This study, conducted in Namibia's Khomas region, aims to assess human factors affecting the HIS and evaluate progress made from 2012 to 2022. It draws on recommendations from a 2012 USAID assessment to provide insights and propose strategies to enhance healthcare delivery and resource allocation.

### Methods and analysis

This study utilizes a cross-sectional design and a multi-method approach to evaluate the performance of the Health Information System (HIS). Qualitative component includes 17 in-depth interviews with key informants, a retrospective document review from the Ministry of Health and Social Services (MoHSS) headquarters in Windhoek, supplemented by a modified office/facility checklist from all 14 health facilities in the Khomas region. The quantitative component involves administering a questionnaire to 330 staff members, using an adapted version of the Performance of Routine Information System Management (PRISM)'s Organizational and Behavioural Assessment Tool (OBAT). Descriptive statistics will be applied to analyse the quantitative data, while a deductive interpretive approach will guide the qualitative data analysis.

**Data availability statement:** No datasets were generated or analysed during the current study. All relevant data from this study will be made available upon study completion.

**Funding:** The author(s) received no specific funding for this work.

**Competing interests:** The authors have declared that no competing interests exist.

## Ethics and dissemination

Ethical approval was obtained from the Stellenbosch University Health Research Ethics Committee (Reference No: S23/05/119) and the Namibia Ministry of Health and Social Services (Reference No: 22/3/2/1). The study will be conducted in accordance to the principles of the Declaration of Helsinki (1964). The study seeks to identify barriers and facilitators for implementing recommendations across different levels of the Health Information System (HIS), with a focus on improving HIS functionality in the Khomas region. Dissemination plans include sharing findings with the study population, presenting at both local and international conferences, and publishing peer-reviewed journals.

## Introduction

Health information is an important building block of the health system and is vital for the effective functioning of the other five building blocks – service delivery, health workforce, medical products, vaccines and technologies, financing as well as leadership and governance [1]. Hence data generated from the health information system (HIS) should be fit for purpose and of good quality, i.e., accurate, reliable, integrated and analysed in a timely fashion [1]. A functional HIS involves a continuous process of information gathering, processing, transmission, storage, and management within the health sector [1–3] and should provide reliable information sources for decision-making especially for resource allocation as well as policy and strategic plannings to ensure the delivery of good healthcare services [3]. Studies show that the quality of a good HIS is affected by the data collection methods and tools, the availability and competency of the staff responsible for collecting and using the information, as well as how and when the information is analysed [1–4].

The World Health Organization (WHO) together with agencies such as the United States Agency for International Development (USAID) and the United States President's Emergency Plan for AIDS Relief (PEPFAR), plays a vital role in supporting developing countries to strengthen their HIS [5–7]. This support includes workforce development through scholarships and training workshops, funding for HIS programmes, provision of equipment to facilitate data collection and analysis at different levels of the HIS, as well as recruitment of qualified personnel, and providing recommendations to improve the HIS [5–7]. Despite this support, low- and middle-income countries (LMICs) continue to face persistent challenges in implementing key recommendations [3,8]. These recommendations focus on clearly defining staff roles and structures, particularly for HIS, providing ongoing training for staff, offering incentives to retain personnel [3,5,7–11].

### Background

Namibia, an upper-middle-income country whose national HIS is led by the Ministry of Health and Social Services (MoHSS), faces challenges similar to other

developing countries [9,10]. These challenges include insufficient HIS staff, absence of HIS policies, poorly defined staff roles especially for the health care workers taking on multiple roles including data capturing and management, as well as the absence or insufficient equipment [2,5,8,10–15]. The most prominent challenges, however, relate to the human component of the HIS, such as shortage of HIS experts and staff to develop the system, unclear organizational structures, limited opportunities for training, reliance on untrained personnel in high-turnover settings [3,10,11,13,15], and resistance to change among staff [15]. Additional barriers include fragmented and non-integrated data systems, lack of monitoring and evaluation mechanisms, insufficient resources to support and sustain HIS interventions, political constraints, weak policy frameworks, and inadequate infrastructures, such as limited access to computers and reliable internet connectivity [3,10,11,13,15].

In 2012, USAID assessed Namibia's national Health Information Systems (NHIS) with the aim of integrating and strengthening the system. The assessment sought to take stock of existing systems, identify strengths and weaknesses, and provide recommendations to guide future planning [11]. Findings were organised into four thematic areas: data and information; technology, protocols, and the human interface; information products, data use, and knowledge management; and management, coordination, and implementation. Key challenges identified included resistance to new systems, inadequate change management strategies, lack of standardized data collection tools, limited managerial capacity, poor work ethics, and shortages of both human and technical resources at local and national levels [11].

The assessment highlighted that most challenges were human-related, with particular emphasis on staff motivation and attitudes [4,16]. It recommended the establishment of dedicated committee within the HIS directorate to oversee strategic planning, coordinate stakeholders, and lead the implementation of recommendations [Top of Form11]. Subsequent studies reported similar challenges, with little evidence of significant improvement [10,13]. An exploratory study conducted six years after the initial assessment in 2012 revealed that the HIS remained fragmented and identified persistent human factor deficits [2], including staff shortages, inadequately trained personnel, and structural and organizational challenges [2].

The main recommendation from a 2012 USAID study regarding the human aspect of the Health Information System (HIS) was the recruitment and training of staff to strengthen HIS capacity [11]. Additional recommendations emphasized the importance of continuous professional development through on-the-job training, refresher courses, and periodic short- and long-term training programmes [11]. However, more than a decade later, there is limited evidence to suggest that these recommendations were implemented or that behavioural changes occurred within the Namibian MoHSS. Recent studies suggest that many of the persistent challenges in the Namibian HIS are largely attributed to human-related factors [2,9–11,14].

A review of related studies conducted in Namibia reveals that many share similar recommendations across technical, behavioural, and organizational domains. Nonetheless, these studies often fall to provide clear evidence of system improvements over time, beyond documenting recurring challenges and the introduction of new information [2,10,11,13,15]. Despite repeated calls to enhance the quality of data generated by the National Health Information System (NHIS), persistent weakness remains in the performance of the Regional Health Information System (RHIS), most of which are attributed to human factors. The extent to which the 2012 USAID recommendations were implemented, and their potential impact on RHIS performance remains uncertain. Furthermore, existing evaluations of the Namibian HIS have rarely examined in depth the human factors influencing system performance [2,10,11,13,15].

The proposed study seeks to address these gaps by applying the Performance of Routine Information System Management (PRISM) framework and tools, specifically the Organizational and Behavioural Assessment Tools (OBAT), in combination with qualitative methods. The study will assess behavioural improvements in the RHIS in Namibia's Khomas region over the past decade and identify barriers and facilitators affecting the implementation of HIS-related recommendations. In doing so, it will evaluate the role of human factors in shaping HIS performance in the Khomas region and determine the extent of progress achieved between 2012 and 2022, drawing upon the recommendations from the 2012 USAID National Health Information Systems Assessment. The aims to address the following research questions:

1. Has there been any improvement in the health information system in Khomas region in terms of human resource capacity (skills and competencies) since 2012?

2. What are the behavioural and organizational factors influencing the performance of health information systems in Khomas region?

3. What are the barriers and facilitators affecting the implementation of recommendations aimed to improve the NHIS?

These questions will be achieved through the following objectives:

1. To assess trends in HIS-related human resources allocations in terms of staff capacity, skills, and competencies in the Khomas region since 2012.

2. To identify the behavioural and organisational factors influencing the performance of the NHIS in the Khomas region.

3. To determine the barriers and facilitators of implementing the recommendations from the USAID assessment of the NHIS in the Khomas region.

## Materials and methods

### Conceptual framework

The PRISM framework is a well-established tool for assessing and improving routine health information systems (RHIS) performance [17]. It examines three key domains: behavioural, organizational, and technical determinants of RHIS. Behavioural determinants focus on staff-related factors, including knowledge, skills, attitudes, and motivation. Organizational determinants address structural and managerial elements, such as resource availability, information culture, and management support. Technical determinants relate to aspects such as the design and availability of data collection tools, processes, and systems [17–23].

The framework comprises four primary tools designed to assess or evaluate RHIS performance. The Performance Diagnostic Tool targets technical determinants, while the Facility or Office Checklist evaluates the relationship between resource availability and RHIS implementation at the facility level, emphasising technical aspects. The Management Assessment Tool (MAT) examines RHIS management practices to enhance overall HIS management. Finally, the Organizational and Behavioural Assessment Tool (OBAT) investigates behavioural and organizational factors that influence RHIS performance [17–21,24–26].

The PRISM framework and its associated tools have been successfully applied to enhance HIS performance in several African countries, including Ethiopia [24], Uganda [21], and South Africa [18]. However, there is no documented evidence of their use in Namibia. Applying the OBAT tool to study behavioural and organizational determinants in the Khomas region offers a unique opportunity to generate context-specific insights and inform strategies to improve RHIS performance locally.

### Study design

The study adopts a cross-sectional design employing a multi-method approach to assess the performance of the HIS in the Khomas region of Namibia. The multi-method approach integrates both qualitative and quantitative methods to provide a comprehensive assessment of HIS performance [27].

The PRISM's OBAT questionnaire and Office/Facility checklist will be adapted and contextualised for Namibia to capture behavioural and organizational factors that promote the effective use of health information [8]. In addition, an interview schedule adapted from a study conducted in Senegal [25] will be used to conduct semi-structured interviews with key informants. The interview schedule is divided into two sections: one targeting the national- and mid-level analyst

and M&E personnel, and the second designed for high-level decision-makers [25]. Key informants will include high-level decision-makers, as well as mid-level analysts involved in HIS implementation and management.

Furthermore, a retrospective document review will be conducted, guided by a structured checklist. Documents to be reviewed will include the HIS policy, latest minutes of HIS committee meetings, latest records on the availability of support structures, such as study leave, loans, or bursaries provided by the MoHSS to improve staff knowledge and capacity.

## Study setting

This study will be conducted in the Khomas region, one of Namibia's 14 administrative regions [28]. The region includes the Windhoek district, which serves as the capital city of Namibia. According to the 2018 situation analysis on human resources for health in Namibia, the Khomas region spans a geographical area of 36,964 km$^2$ and has an estimated population of 447,636. The population-to-health facility ratio in the regions is approximately 37,303:1 [29].

## Health facilities

The Khomas region comprises 14 public health facilities, 11 of which is located in Windhoek and three situated on the outskirts, within 40–110 km of the city. These facilities include one national referral hospital (Windhoek Central Hospital), one intermediate hospital (Intermediate Katutura State Hospital), three health centers (Okuryangava, Khomadal, and Katutura Health Centers), and ten clinics (Wanaheda, Robert Mugabe, Otjomuise, Maxuilili, Hakahana, Donkerhoek, Dordabis, Groot Aub, and Baumgartsbrunn Clinic). The Khomas region was selected as the study site due to its high patient-to-staff ratio, diversity of health facility types, and the presence of the national Health Information and Research Directorate of the MoHSS, which facilitates access to essential data for the study.

## Data collection and analysis

Data will be collected from electronically all the 14 public health facilities in the Khomas region, as well as from the health information systems department at the MoHSS headquarters in Windhoek, using the Stellenbosch University (SUN) survey, a web-based data collection and management system. The principal investigator and two research assistants will collect the data. All staff will undergo intensive training on the OBAT questionnaire, the office/facility checklist, the retrospective document review checklist, and interview schedules. The training will also include interviewing techniques, and accurate recording of responses to ensure data reliability and consistency.

## Study design, data collection and analysis

**Objective 1.** We plan to assess trends in HIS-related human resources allocations in terms of staff capacity, skills, and competencies in all 14 health facilities in the Khomas region since 2012. This comparative and exploratory study will use both qualitative and quantitative methods:

a) Qualitative: Retrospective document review of HIS policies, minutes from HIS committee meetings, and information on study leave, loans, and bursaries.

b) Quantitative: Facility survey of health information personnel and resources in the Khomas region health facilities and the Health Information and Research Directorate (HIRD) at the MoHSS.

A retrospective document review will assess trends in the availability of healthcare workers in selected facilities from 2012 to 2022. This review will include HIS policies, the latest minutes from HIS committee meetings, and information on study leave, loans, and bursaries. The data will be compared across health facilities and against staff establishments (i.e., the total number of approved jobs or posts in an organization) to determine overall improvements. Additional data to be assessed will include the availability of a human resource (HR) information system (manual or electronic), training records

or proof of training, HIS-related policies, committees and directorates, and support structures like study leave, loans, or bursaries provided by the MoHSS to enhance staff knowledge and capacity. These factors will be compared over time to identify any changes.

This document review will be conducted simultaneously with in-depth interviews with HIS department cadres, including the HR department. Information will be collected across different years to evaluate overall improvements in the HIS system. Where historic records are unavailable, the most recent data will be used. In addition, a modified PRISM office/facility checklist will be administered at all 14 health facilities and within the human resource (HR) department to review HR cadre availability and training records over time.

The analysis of the document review will employ a deductive content analysis approach, which facilitates the systematic interpretation of information contained within the documents in order to derive evidence-based conclusions and recommendations. A document review checklist organised into four key categories: 1) the availability of an HR information system (electronic or manual); 2) existing HIS policies and strategic plans; 3) existing HIS committee and directorate; and (4) the availability of support services, such as study leave, loans, and bursaries.

Microsoft Excel will be used as management tool to systematically track and organize the documents under review. The documents will be categorized according to their dates and other relevant metadata. In addition, detailed notes will be compiled to capture observations related to the current structure, functionality, and patterns identified over the years.

Data from the Office/Facility checklist will be exported onto STATA Version 16 and analysed using the ANOVA two way. Dependent and independent variables will be assessed, and graphs such as bar charts or histograms will be generated to assess trends since 2012, in terms of staff turnover, training, experience or education, as well as compare the organizational and behavioural factors. The average absolute deviation from the mean will also be calculated annually to assess data consistency.

**Objective 2.** A comparative and an analytical observational study will be undertaken to examine the behavioural and organisational factors influencing the performance of the NHIS in the Khomas region. Data will be collected through a self-administered questionnaire adapted from the PRISM's Organizational and Behavioural Assessment Tool (OBAT). The questionnaire will be administered to participants from health facilities in the Khomas region and the Health Information and Research Directorate (HIRD) at the MoHSS. It will assess participants' behaviours, attitudes, and knowledge, while also identifying the organizational and behavioural determinants that influence the performance of the NHIS in the Khomas region.

## Sample size and sampling

Sample size estimation was conducted in consultation with a statistician. The target population consisted of 1,513 healthcare workers in the Khomas region, including 1,489 nurses (522 enrolled nurses and 967 registered nurses) and 24 data management staff (15 clerks, 5 HIS officers, and 4 departmental staff).

The minimum required sample was calculated using the formula for an infinite population, where $n$ is the required sample size from unlimited population, $Z$ is the Z-score at the desired confidence level (1.96 for 95%), $p$ is the estimated population proportion (0.5 for maximum variability), and $\varepsilon$ is the margin of error (0.05). Substituting these values produced a sample size of 384 (Fig 1). Applying the finite population correction for N = 1,513 yielded a final sample of approximately 306 nurses. To ensure representativeness, all 24 data management staff were included, resulting in a total target sample of 330 participants.

Power analysis indicated that a minimum of 158 participants would be required to achieve 80% power at a 95% confidence level with a medium effect size (f = 0.25). The planned sample of 330 participants exceeds this threshold, providing sufficient statistical power to detect even smaller effect sizes (f = 0.17).

The 306 healthcare workers will be proportionally drawn from staff registers across all 14 public health facilities, and outreach services. These facilities include those under the Ministry of Health and Social Services (MoHSS), as well as

$$n^* = \frac{n}{1 + \frac{Z_{\frac{\alpha}{2}}^2 * p(1-p)}{\varepsilon^2 * N}}$$

$Z_{\frac{\alpha}{2}}$ is Z- score = 1.96

$\varepsilon$ is margin of error = 5% = 0.05

N is population size = 1489

p is population proportion = 50% = 0.5

n is the sample size from unlimited population.

$$n = \frac{Z_{\frac{\alpha}{2}}^2 * p(1-p)}{\varepsilon^2} = \frac{1.96^2 * 0.5(1-.05)}{0.05^2} = 384.16$$

Therefore

$$n^* = \frac{384.16}{1 + \frac{1.96^2 * 0.5\,(1-0.5)}{0.05^2 * 1489}} = 306 \text{ (rounded up)}$$

**Fig 1. Sample size calculation from a finite population [30–31].**

those funded by external partners such as PEPFAR, I-TECH (International Training and Education Centre for Health), and the Global Fund. Within each facility, systematic sampling with a random starting point will be applied to select participants. If a selected staff member is unavailable, the next eligible individual on the register will be approached until the target number is reached.

The PRISM's OBAT questionnaire will be administered to 330 participants, including 306 nurses, 15 data clerks, five HIS officers, and four staff from the HIS department. Given the small number of HIS staff, all will be included to maximize the information obtained. The questionnaire consists of four sections tailored to different levels of staff: staff and management at all levels, district and higher levels, health facility in-charge, and data management staff. These sections aim to identify organizational and behavioural factors influencing the HIS systems. The results will provide insights into overall improvement.

To ensure data reliability, the researcher will conduct random checks of the captured data. Data will be extracted into Microsoft Excel and compared using the Excel Comparison tool, with any discrepancies validated against the captured questionnaires. Additional data cleaning and validation will involve generating frequency tables for each variable to identify missing values or inconsistencies that may have occurred during data capture.

Following data validation, the dataset will be exported to STATA Version 16 for statical analysis. A two-way ANOVA will be employed to assess the relationship between dependent and independent variables. Multiple linear regression will further be applied to control for potential confounders, thereby allowing for the examination of the effects of professional cadre and facility type on OBAT scores while adjusting for variables such as years of experience in the health system, level of formal HIS training, type of facility (clinic vs. hospital), and highest education attained.

**Objective 3.** This study adopts an explanatory design, relying on qualitative methods of data collection. In particular, semi structured in-depth interviews with key informants will be conducted to explore the barriers and facilitators to implementing the recommendations from the USAID assessment of the NHIS in the Khomas region.

Key informants will include personnel from all 14 public health facilities in the Khomas region, as well as personnel from the MoHSS Health Information Systems. Eligible participants must be stationed at MoHSS health facilities in the Khomas

region, either as MoHSS employees or donor-funded staff. To qualify, respondents must have worked with HIS data at the health facility for a minimum of six months. Eligible cadres include data entry staff, facility in-charges (managers), and HIS managers. Staff employed at private health facilities will be excluded.

Purposive sampling will be used to select 17 participants for the interviews. This sample will comprise one senior nurse or in-charge nurse from each of the 14 health facilities, one senior or lead data clerk, one senior HIS officer, and one staff member from the HIS department.

The semi structured in-depth interviews will be conducted face-to-face or virtually, depending on participants' availability, and guided by an interview schedule. The schedule is divided into two parts: the first, designed for the national-level HIS staff, will be administered to one participant; the second, designed for mid-level analysts and M&E personnel, will be administered to the remaining 16 participants (14 in-charge or senior nurses at each health facility and two district or regional HIS officers). Interviews will last approximately 20–30 minutes, will be voice-recorded, and conducted in English or, where appropriate, in the local languages (Oshindonga and Afrikaans). All recordings will be translated to English prior to analysis. Observational notes will be taken where necessary to complement the in-depth interview data.

## Data saturation

Data saturation will be assessed after all interviews have been transcribed and the key themes thoroughly analysed and identified. This approach ensures that no relevant details are overlooked and that a comprehensive thematic analysis is achieved [32]. By allowing additional time to interpret the data fully, this method identifies all themes, unique insights, and potential outliers. It also minimizes the risk of prematurely concluding that saturation has been reached.

All voice-recorded data will be transcribed and coded using ATLAS.ti. Transcriptions of the interviews conducted in the local languages (Oshindonga and Afrikaans) will be translated into English before analysis. The principal investigator, together with research assistants, will review the transcripts to verify accuracy and completeness, with special attention to ensuring all relevant codes are captured. A deductive thematic approach will then be applied to identify the themes and identify barriers and facilitators to implementing the recommendations from the USAID assessment of the NHIS in the Khomas region. Table 1 presents the alignment between study objectives and data collection tools, while Fig 2 illustrates the study timelines.

## Pilot study

A pilot study will be conducted in a different region, specifically at Okahandja District Hospital and Nau-Aib Clinic, using the same questionnaire and targeting similar participants from hospitals and clinics. The pilot will include 10% of the total sample size, totalling 34 participants, consisting of 31 nurses, one district HIS officer, and two data clerks.

**Table 1. Linkages between the study objectives and data collection methods.**

| Data collection | Objective 1 *(Trends assessment in HIS-related human resources)* | Objective 2 *(Identify behavioural and organisational factors)* | Objective 3 *(Barriers and facilitators)* |
|---|---|---|---|
| **Quantitative methods** | | | |
| Self-administered questionnaires with national and facility level staff (PRISM OBAT) | | X | |
| A modified/office or facility checklist | X | | |
| **Qualitative methods** | | | |
| A retrospective document review (data review on staff trends from the system | X | | |
| In-depth interviews with key informants | | | X |

**Data collection and Analysis Guntt Chart**

| Timeline of data collection and analysis | 2024 | | | | | | | | | 2025 | | | | | | | | | | | |
|---|---|---|---|---|---|---|---|---|---|---|---|---|---|---|---|---|---|---|---|---|---|
| Quarter | Q2 | | | Q3 | | | Q4 | | | Q1 | | | Q2 | | | Q3 | | | Q4 | | |
| Month | Apr | May | Jun | Jul | Aug | Sep | Oct | Nov | Dec | Jan | Feb | Mar | Apr | May | Jun | Jul | Aug | Sep | Oct | Nov | Dec |
| Pilot study | | ▓ | ▓ | | | | | | | | | | | | | | | | | | |
| Data collection: Quantitative and Qualitative | | | | | ▓ | ▓ | ▓ | | | | | | | | | | | | | | |
| Data collection: Document review | | | | | | | | | | | | | | | ▓ | ▓ | | | | | |
| Data analysis | | | | | | | | | | ▓ | ▓ | ▓ | ▓ | ▓ | ▓ | ▓ | ▓ | ▓ | | | |
| Report writing | | | | | | | | | | | | | | | | | | | ▓ | | |
| Final research submission and completion | | | | | | | | | | | | | | | | | | | | | ▓ |

**Fig 2. Gantt chart showing the study timeline.**

The primary purpose of the pilot study is to assess the feasibility and appropriateness of the data collection tools for the main study in the Khomas region. It will assess clarity of the questions, estimate the time required for completion, and allows for adjustments prior to the full fieldwork. This process reduces potential bias, ensures standardization of the data collection instruments, and ensures the reliability of the findings. Any necessary modifications identified during the pilot will be implemented before conducting the main study, thereby confirming that the methods are suitable and capable of yielding high-quality data.

## Ethical considerations

The study obtained ethical clearance from the Stellenbosch University Health Research Ethics Committee (Reference No: S23/05/119) and received permission from the Namibian Ministry of Health and Social Services (Reference No: 22/3/2/1). Both verbal and written informed consent will be obtained from participants of the OBAT survey and in-depth interviews. The consent for the survey will be sought electronically through an introductory page outlining the study details, followed by a consent statement that participants must agree to by clicking a checkbox or button before proceeding. Consent for the in-depth interviews will be obtained verbally, the consent process will be recorded, including explicit permission for audio recording. This ensures participants are aware of their rights, confidentiality measures, and the benefits of participation. Participants are free to decline participation in the survey or interviews without any impact on the study.

Data management will adhere to the Protection of Personal Information Act (POPIA). Data will be stored on a password-protected computer accessible only to the principal investigator. Voice recordings will be destroyed after research completion. These are all in line with the ethical guidelines of the Namibian Ministry of Health and Social Services, as well as the Stellenbosch university ethical guideline, in addition to the POPIA. Respondents and health facilities will be anonymized using unique codes (e.g., HF1 for health facilities, N1 for nurses). Confidential interviews will be conducted in designated private spaces.

## Discussions

### Data dissemination

The results of this study will be disseminated to the MoHSS, including the staff who participated in the research. The principal investigator will be available to present the findings upon request by the MoHSS or other relevant stakeholders.

Furthermore, the study will be submitted for publication in a peer-reviewed journal, and the authors will remain accessible to provide explanations or clarifications of the findings to interested parties.

To protect participant privacy and maintain confidentiality of the Namibian health information system, the datasets generated and analysed during this study will not be publicly shared in a repository. However, access to the data may be granted upon request to the authors, subject to review in accordance with the ethical guidelines of the Namibian Ministry of Health and Social Services.

The inclusion of the PRISM's OBAT tool, retrospective document review, and in-depth interviews strengthens the methodological rigor of this study. Notably, this research is the first in the country to apply two PRISM tools, the OBAT and modified office/ facility tools, to assess and evaluate the HIS system after a decade. The study will provide valuable insights into overall progress and informing the region on the human aspects of the HIS, as well as identify barriers and facilitators influencing the implementation of recommendations across various HIS levels.

However, the generalizability of the findings may be limited to the Khomas region, which has a relatively diverse health-care workforce and comparatively greater resource availability and accessibility, including the presence of the country's reference hospital. Furthermore, the by focusing primarily on the human and behavioural aspect of the HIS, the study may not capture other critical factors, such as the availability of functionality of data collection tools.

## Author contributions

**Conceptualization:** Veronika Jatileni, Edward Nicol.

**Data curation:** Edward Nicol.

**Formal analysis:** Veronika Jatileni, Edward Nicol.

**Investigation:** Edward Nicol.

**Methodology:** Edward Nicol.

**Supervision:** Edward Nicol.

**Validation:** Veronika Jatileni.

**Writing – original draft:** Veronika Jatileni, Edward Nicol.

**Writing – review & editing:** Veronika Jatileni, Edward Nicol.

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
