## [Decision Letter · Decision Letter 0]

21 Oct 2024

Dear Dr. Nicol,

Thank you for submitting your manuscript to PLOS ONE. After careful consideration, we feel that it has merit but does not fully meet PLOS ONE’s publication criteria as it currently stands. Therefore, we invite you to submit a revised version of the manuscript that addresses the points raised during the review process.

We look forward to receiving your revised manuscript.

Kind regards,

Reza Rabiei

Academic Editor

PLOS ONE

Journal Requirements:

2.  During your revisions, please confirm whether the wording in the title is correct and update it in the manuscript file and online submission information if needed. Specifically, this manuscript is a study protocol. Please include the term "Protocol" in the title.

4. Please ensure that you refer to Figure 2 in your text as, if accepted, production will need this reference to link the reader to the figure.

5. We note that Figure 1 in your submission contain map images which may be copyrighted. All PLOS content is published under the Creative Commons Attribution License (CC BY 4.0), which means that the manuscript, images, and Supporting Information files will be freely available online, and any third party is permitted to access, download, copy, distribute, and use these materials in any way, even commercially, with proper attribution. For these reasons, we cannot publish previously copyrighted maps or satellite images created using proprietary data, such as Google software (Google Maps, Street View, and Earth). For more information, see our copyright guidelines: http://journals.plos.org/plosone/s/licenses-and-copyright.

1) You may seek permission from the original copyright holder of Figure 1 to publish the content specifically under the CC BY 4.0 license.  

2) If you are unable to obtain permission from the original copyright holder to publish these figures under the CC BY 4.0 license or if the copyright holder’s requirements are incompatible with the CC BY 4.0 license, please either i) remove the figure or ii) supply a replacement figure that complies with the CC BY 4.0 license. Please check copyright information on all replacement figures and update the figure caption with source information. If applicable, please specify in the figure caption text when a figure is similar but not identical to the original image and is therefore for illustrative purposes only.

Reviewers' comments:

Reviewer's Responses to Questions

**Comments to the Author**

1. Does the manuscript provide a valid rationale for the proposed study, with clearly identified and justified research questions?

Reviewer #1: No

Reviewer #2: Yes

Reviewer #3: No

2. Is the protocol technically sound and planned in a manner that will lead to a meaningful outcome and allow testing the stated hypotheses?

Reviewer #1: Yes

Reviewer #2: Partly

Reviewer #3: No

3. Is the methodology feasible and described in sufficient detail to allow the work to be replicable?

Reviewer #1: Yes

Reviewer #2: Yes

Reviewer #3: Yes

4. Have the authors described where all data underlying the findings will be made available when the study is complete?

Reviewer #1: No

Reviewer #2: Yes

Reviewer #3: Yes

5. Is the manuscript presented in an intelligible fashion and written in standard English?

Reviewer #1: Yes

Reviewer #2: Yes

Reviewer #3: Yes

You may also provide optional suggestions and comments to authors that they might find helpful in planning their study.

Reviewer #1: The manuscript reveals excellent knowledge of the team about study design and scientific data collection to analysis such systems. But I would suggest the authors to submit their manuscript once they have collect and analyzed the information.

There another possibility to change the title and subject of your current manuscript to something new like: Recommendations for HIS assessment in developing countries. Then you will need to rewrite the paper in a way that further details of your recommended methodology and tools become parts of the manuscript.

Reviewer #2: Dear Author

Abstract:

1. The introduction should be shortened.

2. Add discussion.

Manuscript:

1. The discussion should be modified. For this purpose, it is necessary to compare and discuss the stages of the study with other studies.

2. Add the results. In this study, although we have not reached the results yet, it is necessary to briefly write a few lines about when this study will be completed.

And based on the objectives, what results do we expect to achieve at the end of the study.

Resources:

References 11 and 12 should be completed.

Add Availability of data and materials.

Add Implications.

Reviewer #3: Dear Authors,

Thank you for the opportunity to review this manuscript titled “An assessment of the Health Information System in Khomas region, Namibia”. In this manuscript, the authors introduce their study proposal towards research that aims to assess human factors affecting the Health Information System (HIS) in Namibia. Given challenges of a low resource setting, they aim to evaluate progress of the HIS from 2012-2022, against the 2012 USAID recommendations. The topic is essential and of particular interest to both practitioners and policy-makers today as it concerns the exponential implementations of HIS in the health sector. Although intended to better patient management, these implementations come with their own set of challenges.

Overall, the paper makes for a clear and eloquent read, with plans of the study protocol clearly documented. The authors have provided adequate supporting background literature from countries with similar contexts in the region. There is clarity in the chosen framework and findings from previous studies, forming a baseline for comparison in the future. The authors mention three distinct objectives and a detailed methods section on how they plan to achieve these objectives.

The paper, however, requires much work to be considered further for publishing. Some of my suggestions are in keeping with PlosOne’s study protocol guidelines https://journals.plos.org/plosone/s/submission-guidelines#loc-study-protocols. Although this protocol is not for a clinical trial or systematic review, I recommend that the authors refer to the guidelines to enhance the quality of their submission. My feedback and queries for clarification are as follows:

1. Title: It is recommended that the title of the manuscript have the word ‘protocol’ in it (please see guidelines).

2. Background:

a. The protocol claims gaps in literature pertaining to human factors that influence HIS efficiency. It would help the reader to know what authors mean by human factors? Will they be, for example, adopting findings from the cited Nikol et al. 2013 paper that mentions human factors? If not defined before, what is the authors’ working operational definition of human factors? Previous studies, like Kremer et al. 2019 and Hudson, D. 2023, may be considered as example of human factors, for instance.

b. Although aims and objectives have been mentioned, what is the study’s main research question?

3. Materials and Methods:

a. Conceptual framework: Referring to the use of the PRISM framework, it will help to understand whether the study will be a mere replication of the studies done in other countries because there is no prior work done in Namibia? What about the context in Namibia is ‘intriguing’ or demands such a study be replicated? How are findings from this study expected to add value to the contributions already made?

b. Multi-method approach: Please provide a citation for the approach used.

c. The sample size calculation:

• The number of participants calculated for the questionnaire has been demonstrated well. “306 participants will be randomly recruited”. Please elaborate on this. Consider specifying clearly that the sample size calculation was only for the larger number of nurses. The questionnaire would be offered to all of the remaining HIS staff. And why this is the case.

• How was the number of participants for key informant interviews derived or pre-determined to be 17?

d. Data saturation: please mention how data saturation will be determined for the qualitative part of the study.

e. Informed consent: The protocol has mentioned that informed consent will be obtained. How will this be done? (For example, through a patient information sheet, language etc.)

f. Data collection tools: Will the interviews to address Objective 3 be structured, semi-structured, unstructured? What are some topics that will be covered in these? Will the authors be submitting a copy of the checklists, interview guides etc. used for data collection?

g. Data analysis: Outline some potential assumptions and explicitly describe what aspects of the proposed analyses are exploratory. The protocol mentions that a “document review checklist has been developed, which includes four categories”. It will be valuable for the reader to know what these (working) category names are.

4. Pilot study: How will the authors consider modifications, if required, from the feedback during the pilot study?

5. Ethical considerations: Please include a note on consent taking for all participates. How will consent be taken? What contingency steps will be considered if there are refusals?

6. Discussions: Consider enhancing the discussions by addressing: what contributions do the authors expect from their study? Which ongoing discussions and debates around HIS will benefit from your empirical findings? Will there be any contributions towards the conceptual framework?

7. Proof reading suggestions:

a. Citations: In Introduction, revise line 8 citations to read as either “1,3” or “1-3”.

b. Some sentences are densely structured and may require to be revised with adequate connecting words. Wherever possible, these lines are highlighted. Please consider revising these lines for simplicity:

• Introduction – last line; “Despite the assistance… staff to retain them”.

• Background – last line in 1st paragraph; “Other challenges highlighted… computers and internet”.

c. Objective 2: “A quantitative of… (MOHSS). Please revise grammar of sentence.

d. Objective 2: “The sample size calculation depicted in Figure 1”. Please change this to Figure 2 for Sample size calculation. Also, please provide reference for the formula used for the estimation.

e. Acronyms: Please check that acronyms are used after full form presented. For example, MOHSS has been repeated in full form several times.

Once again, thank you for the opportunity to review this paper. I wish the authors all the very best in their research.

References:

• Hudson D. Physician engagement strategies in health information system implementations. Healthc Manage Forum. 2023 Mar;36(2):86-89. doi: 10.1177/08404704221131921. Epub 2022 Oct 31. PMID: 36314071; PMCID: PMC9975817.

• Kremer L, Leeser L, Breil B. Mental Workload Relating Health Information System - A Literature Review. Stud Health Technol Inform. 2019 Sep 3;267:289-296. doi: 10.3233/SHTI190840. PMID: 31483284.

• Nicol E, Bradshaw D, Phillips T, Dudley L. Human Factors Affecting the Quality of Routinely Collected Data in South Africa [Internet]. Ebooks.iospress.nl. 2013. Available from: https://ebooks.iospress.nl/publication/34107

**Do you want your identity to be public for this peer review?** For information about this choice, including consent withdrawal, please see our Privacy Policy

Reviewer #1: **Yes: ** Mansoor Fatehi

Reviewer #2: No

Reviewer #3: No

---

## [Author Response · Author response to Decision Letter 1]

9 Dec 2024

1. Title: It is recommended that the title of the manuscript have the word ‘protocol’ in it (please see guidelines).

Response

The word protocol has been added to the title. The title is “An assessment of the Health Information System in Khomas region, Namibia: Study protocol” (Page 1)

2. Background:

a. The protocol claims gaps in literature pertaining to human factors that influence HIS efficiency. It would help the reader to know what authors mean by human factors? Will they be, for example, adopting findings from the cited Nikol et al. 2013 paper that mentions human factors? If not defined before, what is the authors’ working operational definition of human factors? Previous studies, like Kremer et al. 2019 and Hudson, D. 2023, may be considered as example of human factors, for instance.

Response

The human factors will be partly adopted from the findings cited in Nicol et al, 2013 paper. The other human factors are also adopted from some papers in Namibia; Blodlo & Hamunyela 2017, Kapepo & Yashik, 2018 and Nengomasha et al, 2018.

These human factors are defined and described in the background, paragraph one, second and third sentences, as well as paragraphs 3 and 4.

b. Although aims and objectives have been mentioned, what is the study’s main research question?

Background

b. Thank you for this comment. The research questions are listed below and have been included in the text.

Research Questions

1. Has there been any improvement in the health information system in Khomas region in terms of human resource capacity (skills and competencies) since 2012?

2. What are the behavioral and organizational factors influencing the performance of health information systems in Khomas region?

3. What are the barriers and facilitators affecting the implementation of recommendations aimed to improve the NHIS?

3. Materials and Methods:

a. Conceptual framework: Referring to the use of the PRISM framework, it will help to understand whether the study will be a mere replication of the studies done in other countries because there is no prior work done in Namibia? What about the context in Namibia is ‘intriguing’ or demands such a study be replicated? How are findings from this study expected to add value to the contributions already made?

Response

The extent to which the 2012 recommendations from USAID have been implemented and their impact on RHIS improvement remain unclear. Additionally, most evaluations of the Namibian HIS have not thoroughly investigated the human factors influencing it 2,10-11,13,15. The proposed study aims to address gaps in HIS by utilizing the Performance of Routine Information System Management (PRISM)’s Organizational and Behavioural Assessment Tools (OBAT) alongside qualitative methods.

How it will add value to the contributions is already addressed under Discussions, data dissemination, paragraph two.

b. Multi-method approach: Please provide a citation for the approach used.

Response

Citation for the Multi method approach added.

c. The sample size calculation:

1. The number of participants calculated for the questionnaire has been demonstrated well. “306 participants will be randomly recruited”. Please elaborate on this. Consider specifying clearly that the sample size calculation was only for the larger number of nurses. The questionnaire would be offered to all of the remaining HIS staff. And why this is the case.

Response

Thank you for this comment. The following paragraph has been added to provide more clarity on the selection process:

“Using the staff registers obtained from the district office, the 306 health workers will be proportionally drawn from each of the 14 health facilities included in the study. Within each facility, a systematic sampling technique with a random starting point will be applied to select the required number of healthcare workers. Additionally, all 24 data management staff will be included in the study. If a selected staff member is unavailable, the research team will proceed to the next available individual on the list, continuing this process until the target number is achieved.

2. How was the number of participants for key informant interviews derived or pre-determined to be 17?

Response

Our choice of 17 participants was based on the following studies that show that code and meaning saturations can be reached with between 16 and 24 interviews.

Hennink MM, Kaiser BN, Marconi VC. Code saturation versus meaning saturation: how many interviews are enough? Qual Health Res. 2017;27(4):591–608.

Guest B, Bunce A, Johnson L. How many interviews are enough? An experiment with data saturation and variablility. Field Methods. 2006;18(1):59–82.

d. Data saturation: please mention how data saturation will be determined for the qualitative part of the study.

Response

Thanks. A paragraph on data saturation has been included in the text. (page 15)

e. Informed consent: The protocol has mentioned that informed consent will be obtained. How will this be done? (For example, through a patient information sheet, language etc.)

Response:

Highlighted under Ethical considerations: Informed consent, both verbal and written, will be obtained from participants of the OBAT survey and in-depth interviews. For the survey, consent will be sought electronically through an introductory page outlining the study details, followed by a consent statement that participants must agree to by clicking a checkbox or button before proceeding. Consent for the in-depth interviews will be obtained verbally, the consent process will be recorded, including explicit permission for audio recording. (Page 16)

f. Data collection tools: Will the interviews to address Objective 3 be structured, semi-structured, unstructured? What are some topics that will be covered in these? Will the authors be submitting a copy of the checklists, interview guides etc. used for data collection?

Response:

Addressed under objective 3: The Semi structured in-depth interviews will be conducted face-to-face or virtually with 17 key informants from health facilities and the Health Information and Research Directorate using an interview schedule.

The document review checklist and interview guide, as well as the links to the OBAT survey and modified facility checklist will be shared by the authors on request.

g. Data analysis: Outline some potential assumptions and explicitly describe what aspects of the proposed analyses are exploratory. The protocol mentions that a “document review checklist has been developed, which includes four categories”. It will be valuable for the reader to know what these (working) category names are.

Response:

Thank you for the comment. This paragraph has been added to the text.

“These categories are as follows: availability of an HR information system (electronic or manual), HIS policies and strategic plans, HIS committee and directorate, and the availability of support services such as study leave, loans, and bursaries.” (page 11)

4. Pilot study: How will the authors consider modifications, if required, from the feedback during the pilot study?

Response:

The purpose of the pilot is to test the suitability of the questionnaires and identify whether further adjustments are needed. The pilot phase will also help in gauging the time required to complete the questionnaire.

After the pilot, revisions will be made to the questionnaire where necessary, and lessons learnt will be adapted accordingly.

See text on page 16 regarding the amendment.

5. Ethical considerations: Please include a note on consent taking for all participates. How will consent be taken? What contingency steps will be considered if there are refusals?

Response:

Thanks. This has already been addressed under the ethical considerations paragraph: (page 16)

6. Discussions: Consider enhancing the discussions by addressing: what contributions do the authors expect from their study? Which ongoing discussions and debates around HIS will benefit from your empirical findings? Will there be any contributions towards the conceptual framework?

Response:

Thank you for your comment. This manuscript is a protocol for a proposed study. The discussions will be more detailed after data collection analysis have been completed and results available.

7. Proof reading suggestions:

a. Citations: In Introduction, revise line 8 citations to read as either “1,3” or “1-3”.

Response:

Thank you

a. rectified to 1-3 (page 4)

b. Some sentences are densely structured and may require to be revised with adequate connecting words. Wherever possible, these lines are highlighted. Please consider revising these lines for simplicity:

• Introduction – last line; “Despite the assistance… staff to retain them”.

Response:

the introduction has been revised.

“Despite support from these agencies, low- and middle-income countries (LMICs) still face challenges in implementing key recommendations. These recommendations focus on clearly defining staff roles and structures, particularly for Health Information Systems (HIS), providing ongoing training for staff, and offering incentives to retain personnel.” (page4)

• Background – last line in 1st paragraph; “Other challenges highlighted… computers and internet”.

Response:

Background- sentence revised

“Other issues include unintegrated data systems, lack of system monitoring and evaluation, limited resources to support and sustain implemented systems, political barriers, absence of policies; and inadequate infrastructures such as computers and internet” (page 5)

c. Objective 2: “A quantitative of… (MOHSS). Please revise grammar of sentence.

Response:

Thank you. The sentence has been reworded accordingly as follows:

“Data will be collected quantitatively using a self-administered questionnaire adapted from the PRISM’s Organizational and Behavioural Assessment Tool (OBAT). The questionnaire will be administered to participants from health facilities in the Khomas region and the Health Information and Research Directorate (HIRD) at the MoHSS. It will assess participants’ behaviours, attitudes, and knowledge, as well as identify factors influencing the performance of the NHIS in the Khomas region.” (page 11, 12)

d. Objective 2: “The sample size calculation depicted in Figure 1”. Please change this to Figure 2 for Sample size calculation. Also, please provide reference for the formula used for the estimation.

Response:

Addressed. Figure 1 changed to figure 2. Two references for the sample size have been inserted. (pages 12 and 21)

e. Acronyms: Please check that acronyms are used after full form presented. For example, MOHSS has been repeated in full form several times.

Response

Acronyms have been addressed as suggested

---

## [Decision Letter · Decision Letter 1]

25 Feb 2025

Dear Dr. Nicol,

Thank you for submitting your manuscript to PLOS ONE. After careful consideration, we feel that it has merit but does not fully meet PLOS ONE’s publication criteria as it currently stands. Therefore, we invite you to submit a revised version of the manuscript that addresses the points raised during the review process.

We look forward to receiving your revised manuscript.

Kind regards,

Reza Rabiei

Academic Editor

PLOS ONE

Journal Requirements:

Reviewers' comments:

Reviewer's Responses to Questions

**Comments to the Author**

1. Does the manuscript provide a valid rationale for the proposed study, with clearly identified and justified research questions?

Reviewer #3: Yes

Reviewer #4: Yes

2. Is the protocol technically sound and planned in a manner that will lead to a meaningful outcome and allow testing the stated hypotheses?

Reviewer #3: Yes

Reviewer #4: Yes

3. Is the methodology feasible and described in sufficient detail to allow the work to be replicable?

Reviewer #3: Yes

Reviewer #4: Yes

4. Have the authors described where all data underlying the findings will be made available when the study is complete?

Reviewer #3: Yes

Reviewer #4: Yes

5. Is the manuscript presented in an intelligible fashion and written in standard English?

Reviewer #3: Yes

Reviewer #4: Yes

You may also provide optional suggestions and comments to authors that they might find helpful in planning their study.

Reviewer #3: Dear Authors,

Thank you for the second opportunity to review this manuscript now titled “An assessment of the Health Information System in Khomas region, Namibia: Study Protocol”.

I am satisfied with the responses to my review comments and amendments made to the paper before resubmission.

A very minor modification to the manuscript required is to expand the acronym HR in the Introduction.

I refer back to the editor for the final decision regarding this paper and wish the authors all the very best in the next stage of their research.

Reviewer #4: Overall, this manuscript presents valuable insights into healthcare professionals' intentions regarding digital health data hubs in Ethiopia. The topic of digital health data hubs is highly relevant, especially in the context of achieving Sustainable Development Goals (SDGs). Your focus on Ethiopia adds an important dimension to the global conversation about digital health. These merits contribute to making your manuscript a valuable addition to the literature on digital health adoption among healthcare professionals, particularly in developing countries like Ethiopia.

General Feedback

1. Clarity and Conciseness:

a. The abstract is informative but could be more concise. Aim to summarize key findings in fewer words while maintaining clarity

b. Some sentences are complex, consider breaking them into shorter sentences for better readability.

Example from the introduction

Original: "Gradually, more and more people are using mobile devices and the Internet and intend to use them for healthcare services, as digital health (DH) has significantly accelerated the achievement of the Sustainable Development Goals (SDGs) and strengthened healthcare systems in Africa."

Suggested Revision: "More people are gradually using mobile devices and the Internet for healthcare services. Digital health (DH) has significantly accelerated progress toward achieving the Sustainable Development Goals (SDGs) and strengthening healthcare systems in Africa."

Example from Background Section:

Original: "To date, it is of great importance to examine the emerging cloud-based adoption at the organizational level Healthcare organizations have continually recorded data over time for customers, suppliers, and stakeholders to analyze the data and derive insights."

Suggested Revision: "It is important to examine emerging cloud-based adoption at the organizational level. Healthcare organizations have continually recorded data over time for customers, suppliers, and stakeholders to analyze this information and derive insights."

2. Structure:

a. Ensure that each section flows logically into the next. For example, the transition from the introduction to methods could be smoother by summarizing how the background leads to your research questions or hypotheses.

b. Consider using subheadings within sections (e.g., "Methods," "Results") to improve navigation through the document.

3. Methodology:

a. It might be helpful to provide more detail about how you ensured the validity and reliability of your survey instruments beyond mentioning pre-testing and PCA.

b. Clarify how you addressed potential biases in sampling or data collection.

4. Results Presentation:

a. Present results in a clear manner using tables or figures where appropriate (e.g., showing demographic data).

5. Discussion:

a. The discussion should connect back to your research questions/hypotheses more explicitly.

b. Consider discussing limitations earlier in this section rather than at the end; this can help contextualize your findings.

6. Conclusion:

Your conclusion summarizes findings well but could benefit from a stronger emphasis on implications for practice and future research directions.

Specific Feedback

Abstract:

a. The phrase "the health system has a unified digital health center" might need clarification - does it mean there is one central hub for all data?

b. Instead of stating "this study aims," use past tense since this is a completed study. "This study assessed..."

Introduction

a. The introduction provides good context but could be streamlined by focusing on key points relevant to your study's objectives.

b. Avoid excessive citations in introductory paragraphs; instead, synthesize information from multiple sources into cohesive statements.

Methods

a. Specify what type of healthcare professionals were surveyed (e.g., doctors, nurses) early on for context.

b. In describing SEM analysis, briefly explain why this method was chosen over others.

Results

a. Include descriptive statistics before diving into inferential statistics to give readers context about your sample.

b. Be cautious with terms like “significant” without specifying p-values initially; clarify what constitutes significance based on your analysis plan.

Discussion

a. Expand on how these findings relate to existing literature—what do they add or challenge?

b. Discuss practical applications of your findings more thoroughly—how can stakeholders implement changes based on this research?

Congratulations to the team on this great achievement.

**Do you want your identity to be public for this peer review?** For information about this choice, including consent withdrawal, please see our Privacy Policy

Reviewer #3: No

Reviewer #4: No

---

## [Author Response · Author response to Decision Letter 2]

26 Feb 2025

Reviewer #3:

1. Dear Authors,

Thank you for the second opportunity to review this manuscript now titled “An assessment of the Health Information System in Khomas region, Namibia: Study Protocol”.

I am satisfied with the responses to my review comments and amendments made to the paper before resubmission.

A very minor modification to the manuscript required is to expand the acronym HR in the Introduction.

I refer back to the editor for the final decision regarding this paper and wish the authors all the very best in the next stage of their research.

Response

Thank you. The acronym HR has been written in full as suggested

Reviewer #4

It appears there may have been a mix-up with the comments from Reviewer #4. These comments and feedback seem to pertain to a different manuscript, specifically, a study conducted in Ethiopia, rather than our submission, "An Assessment of the Health Information System in Khomas Region, Namibia: Study Protocol." The comments do not appear to be relevant to our study.

---

## [Decision Letter · Decision Letter 2]

18 Jul 2025

Dear Dr. Nicol,

Thank you for submitting your manuscript to PLOS ONE. After careful consideration, we feel that it has merit but does not fully meet PLOS ONE’s publication criteria as it currently stands. Therefore, we invite you to submit a revised version of the manuscript that addresses the points raised during the review process.

We look forward to receiving your revised manuscript.

Kind regards,

Mohammad Hosein Hayavi-Haghighi, Ph.D

Guest Editor

PLOS ONE

Journal Requirements:

Reviewers' comments:

Reviewer's Responses to Questions

**Comments to the Author**

1. Does the manuscript provide a valid rationale for the proposed study, with clearly identified and justified research questions?

Reviewer #3: Yes

Reviewer #4: Partly

2. Is the protocol technically sound and planned in a manner that will lead to a meaningful outcome and allow testing the stated hypotheses?

Reviewer #3: Yes

Reviewer #4: Partly

3. Is the methodology feasible and described in sufficient detail to allow the work to be replicable?

Reviewer #3: Yes

Reviewer #4: Yes

4. Have the authors described where all data underlying the findings will be made available when the study is complete?

Reviewer #3: Yes

Reviewer #4: Yes

5. Is the manuscript presented in an intelligible fashion and written in standard English?

Reviewer #3: Yes

Reviewer #4: No

You may also provide optional suggestions and comments to authors that they might find helpful in planning their study.

Reviewer #3: Dear Editor,

Thank you for this third opportunity to review this manuscript now titled “An assessment of the Health Information System in Khomas region, Namibia: Study Protocol”.

I have no new comments to make. A very minor modification to expand an acronym in the manuscript was required from my end. The authors have fulfilled this.

However, a matter of concern may be the authors’ statement that the comments from Reviewer 4 do not match with their study topic.

Not much more from my side. I wish the authors all the very best.

Many thanks again for the opportunity.

Reviewer #4: Thank you for submitting this well‑conceived study protocol assessing human and organizational determinants of HIS performance in Namibia’s Khomas region. The strengths include the use of the internationally recognized PRISM framework, a multi‑method approach combining qualitative and quantitative data, and clear alignment of objectives with data collection tools.

Major points to address prior to publication:

1. Explicit Research Questions & Hypotheses: Translate each objective into one or more focused research questions or hypotheses. State whether analyses are confirmatory or exploratory, and align hypotheses accordingly.

2. Statistical Power & Analytical Plan: Provide a formal power calculation (e.g., minimum detectable effect size at α = 0.05, 80% power), or justify the adequacy of n = 330 for planned analyses. Describe how potential confounders will be controlled for in the ANOVA or through additional covariate analyses.

3. Data Sharing Details: Specify the target public repository (with DOI if known) and the types of data files to be deposited (e.g., .csv, .sav for survey data; transcripts or codebooks for qualitative data).

4. Manuscript Editing: Engage a professional copy‑editor to correct grammatical errors and improve flow. Ensure all figures and tables are properly embedded with visible captions.

Minor suggestions:

- Clarify the timeline of data collection and analysis in a brief Gantt chart or timeline paragraph.

- Define any acronyms upon first use (e.g., “M&E”).

- In the Ethics section, specify how storage and destruction of audio recordings comply with local data protection laws beyond POPIA.

- Repetition & Formatting: The Data Availability question is duplicated in the submission form.

- Grammar & Style: Instances of missing articles (“A functional HIS involve…” should be “A functional HIS involves…”), subject–verb disagreement (“These aids come in the forms…” → “These aids come in the form…”), and inconsistent tense.

- Typos & References: Reference numbering sometimes jumps (e.g., “1.3”), and spacing around citations is inconsistent. Figure captions refer to “Figure 1” but the embedded image link is not visible in the manuscript PDF.

- Clarity: Long paragraphs (especially in the Introduction) should be broken up and streamlined.

Recommendation: A careful copy‑edit is needed to correct grammatical errors, ensure consistent referencing, and improve overall readability before peer review.

**Do you want your identity to be public for this peer review?** For information about this choice, including consent withdrawal, please see our Privacy Policy

Reviewer #3: **Yes: ** Carolyn Kavita Tauro

Reviewer #4: No

---

## [Author Response · Author response to Decision Letter 3]

4 Sep 2025

REVIEWER #3

Thank you for this third opportunity to review this manuscript now titled “An assessment of the Health Information System in Khomas region, Namibia: Study Protocol”. I have no new comments to make. A very minor modification to expand an acronym in the manuscript was required from my end. The authors have fulfilled this.

However, a matter of concern may be the authors’ statement that the comments from Reviewer 4 do not match with their study topic.

Not much more from my side. I wish the authors all the very best. Many thanks again for the opportunity.

RESPONSE:

Thank you for the compliment and the constructive comments.

REVIEWER #4

Thank you for submitting this well‑conceived study protocol assessing human and organizational determinants of HIS performance in Namibia’s Khomas region. The strengths include the use of the internationally recognized PRISM framework, a multi‑method approach combining qualitative and quantitative data, and clear alignment of objectives with data collection tools.

RESPONSE:

Thank you.

Major points to address prior to publication:

1. Explicit Research Questions & Hypotheses: Translate each objective into one or more focused research questions or hypotheses. State whether analyses are confirmatory or exploratory, and align hypotheses accordingly.

RESPONSE:

Thank you for your comment, however, the research questions were already included in the revised submission (see page 6). The research questions are both confirmatory and exploratory, aiming to identify if there has been an improvement since 2012 in terms of the human capacity in the Khomas region. And also, to explore behavioural and organisational factors influencing the performance of the HIS system. These are also highlighted on pages 10, 11, and 14.

2. Statistical Power & Analytical Plan: Provide a formal power calculation (e.g., minimum detectable effect size at α = 0.05, 80% power), or justify the adequacy of n = 330 for planned analyses.

RESPONSE:

The following paragraphs have been included to justify the sample size calculation:

“Sample size estimation was conducted with statistical consultation. The target population comprised 1,513 healthcare workers in the Khomas region, including 1,489 nurses (522 enrolled nurses and 967 registered nurses) and 24 data management staff (15 clerks, 5 HIS officers, and 4 departmental staff). The minimum required sample was calculated using the formula for an infinite population where ‘n’ is the required sample size, ‘Z’ is the z-score at the desired confidence level (1.96 for 95%), 𝑝 is the estimated proportion (0.5 for maximum variability), and 𝑒 is the margin of error (0.05). Substituting these values yields a required sample of 384 (Figure 1). The finite population correction was then applied where N=1,513. This adjustment produced a final sample size of approximately 306 nurses. To ensure representativeness, all 24 data management staff were included, resulting in a total of 330 participants.”

3. Describe how potential confounders will be controlled for in the ANOVA or through additional covariate analyses.

RESPONSE:

The following paragraphs have been included to the analysis section:

“This will enable the assessment of effects of professional cadre and facility type on OBAT scores while adjusting for potential confounders such as years of experience in the health system, level of formal training in HIS, type of facility (clinic vs. hospital) and Highest education attained.”

4. Data Sharing Details: Specify the target public repository (with DOI if known) and the types of data files to be deposited (e.g., .csv, .sav for survey data; transcripts or codebooks for qualitative data).

RESPONSE:

The reprint of this study will be made available at: doi: https://doi.org/10.1101/2024.06.28.24309648

Unfortunately, the datasets generated and analysed during this study will not be publicly shared in a repository in order to protect participants’ privacy and maintain the confidentiality of the Namibian health information system. However, data access will be considered upon a request to the authors. The requests will be reviewed in accordance the ethics of the Namibian Ministry of Health and Social Services.

Manuscript Editing: Engage a professional copy‑editor to correct grammatical errors and improve flow. Ensure all figures and tables are properly embedded with visible captions.

RESPONSE:

Thank you for this suggestion. The manuscript has been edited accordingly

Minor suggestions:

- Clarify the timeline of data collection and analysis in a brief Gantt chart or timeline paragraph.

RESPONSE:

Thank you. A Gantt chart has been included as an appendix as suggested

- Define any acronyms upon first use (e.g., “M&E”).

RESPONSE:

Addressed

- In the Ethics section, specify how storage and destruction of audio recordings comply with local data protection laws beyond POPIA.

RESPONSE:

c) All audio recordings are stored on a password protected laptop; and they will be permanently deleted upon completion of the research. These are all in line with the ethical guidelines of the Namibian Ministry of Health and Social Services, as well as the Stellenbosch university ethical guideline, in addition to the POPIA.

RESPONSE:

- Repetition & Formatting: The Data Availability question is duplicated in the submission form.

Addressed

- Grammar & Style: Instances of missing articles (“A functional HIS involve…” should be “A functional HIS involves…”), subject–verb disagreement (“These aids come in the forms…” → “These aids come in the form…”), and inconsistent tense.

RESPONSE:

Addressed

- Typos & References: Reference numbering sometimes jumps (e.g., “1.3”), and spacing around citations is inconsistent. Figure captions refer to “Figure 1” but the embedded image link is not visible in the manuscript PDF.

RESPONSE:

Addressed

- Clarity: Long paragraphs (especially in the Introduction) should be broken up and streamlined

References for “other models” have been included as suggested

RESPONSE:

Addressed

Recommendation: A careful copy‑edit is needed to correct grammatical errors, ensure consistent referencing, and improve overall readability before peer review.

RESPONSE:

Done

---

## [Editor Report · Decision Letter 3]

10 Sep 2025

A protocol for an assessment of the Health Information System in Khomas region, Namibia

PONE-D-24-25668R3

Dear Dr. Nicol,

We’re pleased to inform you that your manuscript has been judged scientifically suitable for publication and will be formally accepted for publication once it meets all outstanding technical requirements.

Kind regards,

Mohammad Hosein Hayavi-Haghighi, Ph.D

Guest Editor

PLOS ONE
---

## [Editor Report · Acceptance letter]

PONE-D-24-25668R3

PLOS ONE

Dear Dr. Nicol,

I'm pleased to inform you that your manuscript has been deemed suitable for publication in PLOS ONE. Congratulations! Your manuscript is now being handed over to our production team.

Kind regards,

on behalf of

Dr. Mohammad Hosein Hayavi-Haghighi

Guest Editor

PLOS ONE